# The use of paracetamol during pregnancy: A qualitative study and possible strategies for a clinical trial

Cathrine Vedel[1]*, Ditte Staub Jørgensen[1], David Møbjerg Kristensen[2,3,4], Olav Bjørn Petersen[1,5], Gorm Greisen[6]

1 Center of Fetal Medicine and Pregnancy, Department of Obstetrics, Copenhagen University Hospital Rigshospitalet, Copenhagen, Denmark, 2 Department of Neurology, Danish Headache Center, Copenhagen University Hospital Rigshospitalet, Copenhagen, Denmark, 3 Univ Rennes, Inserm, EHESP, Irset (Institut de recherche en santé, environnement et travail) UMR_S, Rennes, France, 4 Department of Biology, University of Copenhagen, Copenhagen, Denmark, 5 Department of Clinical Medicine, Faculty of Health and Medical Sciences, University of Copenhagen, Copenhagen, Denmark, 6 Neonatal Department, Copenhagen University Hospital Rigshospitalet, Copenhagen, Denmark

* cathrinevedel@gmail.com

**Data Availability Statement:** All relevant data are within the paper and its Supporting Information files.

## Abstract

Paracetamol (N-acetyl-p-aminophenol (APAP), also known as acetaminophen) is used to relieve mild to moderate pain and reduce fever. APAP is widely used during pregnancy as it is considered safe when used as directed by regulatory authorities. However, a significant amount of epidemiological and experimental research suggests that prenatal exposure potentially alters fetal development. In this paper, we summarize the potentially harmful adverse effects of APAP and the limitations of the current evidence. It highlights the urgent need for a clinical trial, and the aim of the presented qualitative pilot study on APAP use during pregnancy is the feasibility of a large-scale randomized controlled trial (RCT). In the qualitative study, we included 232 Danish women from three hospitals in the spring of 2021. After recognizing the pregnancy, 48% had taken any APAP, and 6% had taken it weekly or more than weekly. A total of 27% who had taken APAP in the first trimester of pregnancy (even rarely) would potentially participate in an RCT. In a potential clinical trial, the women would need to be included early in the 1st trimester as the suspected harmful effects of APAP lies within this early reproductive developmental window. A possible recruitment strategy was explored. These data suggest that the target population appears positive towards an RCT. As a negative attitude among users has been considered the major hindrance for such a study, we cannot see hindrances for performing an RCT.

## Introduction

Paracetamol (N-acetyl-para-aminophenol, APAP also known as acetaminophen) is a commonly used drug to relieve pain and reduce fever. The medication is considered safe when administered accordingly, is inexpensive, does not require a prescription, and is sold outside

**Funding:** The authors received no specific funding for this work.

**Competing interests:** The authors have declared that no competing interests exist.

the pharmacies in many countries. APAP is used by pregnant women, as both the US Food and Drug Administration (FDA) and European Medicines Agency (EMA) consider APAP safe to use during pregnancy. However, a significant amount of research suggests that prenatal exposure may alter some parts of fetal development leading to urogenital, neurodevelopmental, and reproductive disorders [1–3]. This paper aimed to investigate the feasibility of executing a randomized clinical trial (RCT) in the light of the results of a new qualitative study on APAP use during pregnancy.

## The possible adverse effects of prenatal APAP exposure

APAP freely passages both the placenta and the blood-brain barrier, and a growing body of evidence suggests that it is a potential endocrine disruptor. In a recent consensus statement calling for action on this matter, a thorough overview of the existing literature has been provided [4]. Increasing evidence from both experimental and epidemiological studies suggests that prenatal APAP exposure is associated with male urogenital and reproductive tract abnormalities through anti-androgenic mechanisms. The fact that APAP is a less potent anti-androgen than other pharmaceuticals (for example ketoconazole) means that large studies are needed to be sufficiently powered to detect these effects [4].

## The limitations of the evidence

The epidemiological evidence is limited mainly by confounding by indication, i.e. whether the conditions for which APAP is taken, are the true causes, such as febrile viral illness. Also, self-selection may play a role, i.e. women who decide to use APAP for pain or fever may differ in their choices regarding other aspects of life that may carry risks. There are limited means of adjusting for these sources of confounding since limited relevant data are available in large observational datasets, such as routine health care registries, and non-randomized studies should always be interpreted with caution when used for healthcare decisions [5]. The relevance of the animal studies depends on the assumption of similarity of biology across species which can be inappropriate [6]. Even studies on human cell and organotypic cultures may have limited relevance for the *in vivo* situation, due to limitations in comparability of the dosing of the substance that is tested, and the limitations of compensatory mechanisms in the simpler systems [7].

## Recommendations by government agencies

When evaluating the use of medicine during pregnancy it involves a benefit versus risk assessment, where the potential benefits to mother and fetus are compared to possible risks to the fetus. APAP use during the entire pregnancy is safe to use according to the FDA and the EMA, as they both state that animal studies have failed to demonstrate risks of congenital malformations from prenatal exposure. However, this is not in line with the previously mentioned consensus statement. Moreover, they state that no adequate studies have been performed on pregnant women [8]. Furthermore, it is difficult to point to other, safer medications.

## A need for better evidence

While the recommendations of the medicine agencies may represent a reasonable decision on the best current use of APAP during pregnancy, there is a need for better evidence for better-founded decisions in the future. The gold standard for evidence of the benefits and harms of medications is a randomized clinical trial. The question is if a randomized clinical trial in pregnant women can realistically be performed.

## The requirements for a randomized clinical trial that may be decisive

There are several issues to be considered. First, would a trial in pregnant women, where the investigated drug (APAP) is already thought to have some harmful effect on the fetus be ethical? The solution could be a placebo-controlled, double-blinded, parallel-group, RCT, where the women are only asked to use APAP/placebo in situations where they normally would have used APAP. It is possible that maternal hyperthermia (fever) may cause fetal malformations; hence the use of APAP in cases of fever during pregnancy may decrease the risk of certain malformations, though the evidence is not clear [9, 10]. Consequently, there is a possible risk of both harm and benefit when using APAP during pregnancy.

Second, what would be a patient-relevant primary outcome? Cryptorchidism in boys is a candidate. Although cryptorchidism can be corrected surgically, it remains an intervention-requiring malformation, and surgery does not completely remove the effects on fertility or the risk of cancer. Besides cryptorchidism as a primary outcome, increasing experimental and epidemiological research suggest that prenatal exposure to APAP might also increase the risks of the fellow urogenital disorder hypospadia and adverse neurodevelopmental and behavioral outcomes, such as attention deficit hyperactivity disorder (ADHD), autism spectrum disorder, language delay, and decreased intelligence quotient.

When in pregnancy must a trial be done? Since the male reproductive development occurs in the first trimester of pregnancy, a trial would need to recruit pregnant women already soon after recognition of pregnancy. In Denmark, all pregnant women are offered three visits to their family doctor and five to seven visits with a midwife. Moreover, all women are offered two prenatal ultrasound scans free of charge (taxpaid)–one in the first trimester including risk assessment for aneuploidies and one in the second trimester including assessment of fetal anatomy and possible malformations. More than 97% of the Danish pregnant population attends the prenatal ultrasound screenings [11]. However, enrollment would need to take place before the first-trimester ultrasound scan due to the reproductive developmental window. Hence, possible strategies for earlier enrollment are needed. As women in Denmark often search for information regarding pregnancy and childbirth before conception, study information by social media might be an option, and other studies and trials have already been successful with this kind of recruitment [12–14]. The study should include all women regardless of APAP use earlier in pregnancy; however, early use of APAP should be registered upon inclusion.

What size should a trial have to be decisive? The serious problems caused by APAP in pregnancy must be rare; if not, they had been discovered long ago. Therefore, such a trial must be large. Approximately 2% of Danish boys have a diagnosis of cryptorchidism in the National Patient Register [15]. The epidemiological evidence suggests that the relative risk is about 1.5, corresponding to an absolute risk of 1% and a number needed to harm 100 [15–17]. To detect this with a power of 90%, 5,500 boys in each group are needed. Thus 22,000 women must be randomized. The children must be followed up to two years of age to detect cryptorchidism.

## A qualitative study on APAP use during pregnancy and the acceptability of a randomized trial

To qualify whether a clinical trial would be feasible, we conducted a qualitative study based on a short questionnaire. The study was conducted at the Fetal Medicine Departments at Rigshospitalet, Herlev Hospital, and Slagelse Hospital in the spring of 2021, and we aimed to include 200 completed questionnaires. The study did not need approval by the ethics committee, as it was a questionnaire. The questionnaire was voluntary and filled out anonymously after receiving written information about the study. The questions included:

1. How far along are you in the pregnancy?

2. How often did you use APAP before your pregnancy?

3. How often have you used APAP after realizing that you were pregnant?

4. Would you consider participating in a study, where you would be picked to either take APAP or placebo at times, where you would normally take APAP?

5. Would you participate for free or would it require some kind of dispensation/present?

6. How could we reach you very early in pregnancy?

A total of 232 pregnant women completed the questionnaire with a mean gestational age of 13 weeks and 6 days. A total of 48% had taken APAP after recognizing the pregnancy, and only 6% had taken APAP weekly or more than weekly (Fig 1). The raw results can be found in S1 Table. Overall, 34% would participate in an RCT regarding APAP and 49% would not. The rest were in doubt. Of the women, who had taken APAP after becoming pregnant (rarely, monthly, weekly, more than weekly), 44% would not participate in an RCT. That leaves a total of 27% who had taken any APAP in the first trimester of pregnancy that would potentially participate in an RCT. The total number of births per year in Denmark is 60,000, so potentially 16,000 could be randomized per year. It is common to apply a factor of 0.5 when planning randomized trials, so 8,000 is more realistic, suggesting that a trial could be completed in 2.5 years.

However, this calculation assumes that 'any use of APAP' is associated with a relative risk of 1.5, which may not be true. If we assume that a weekly or more than weekly use of APAP during the first trimester is needed for a relative risk of 1.5, the calculation looks like this: of those

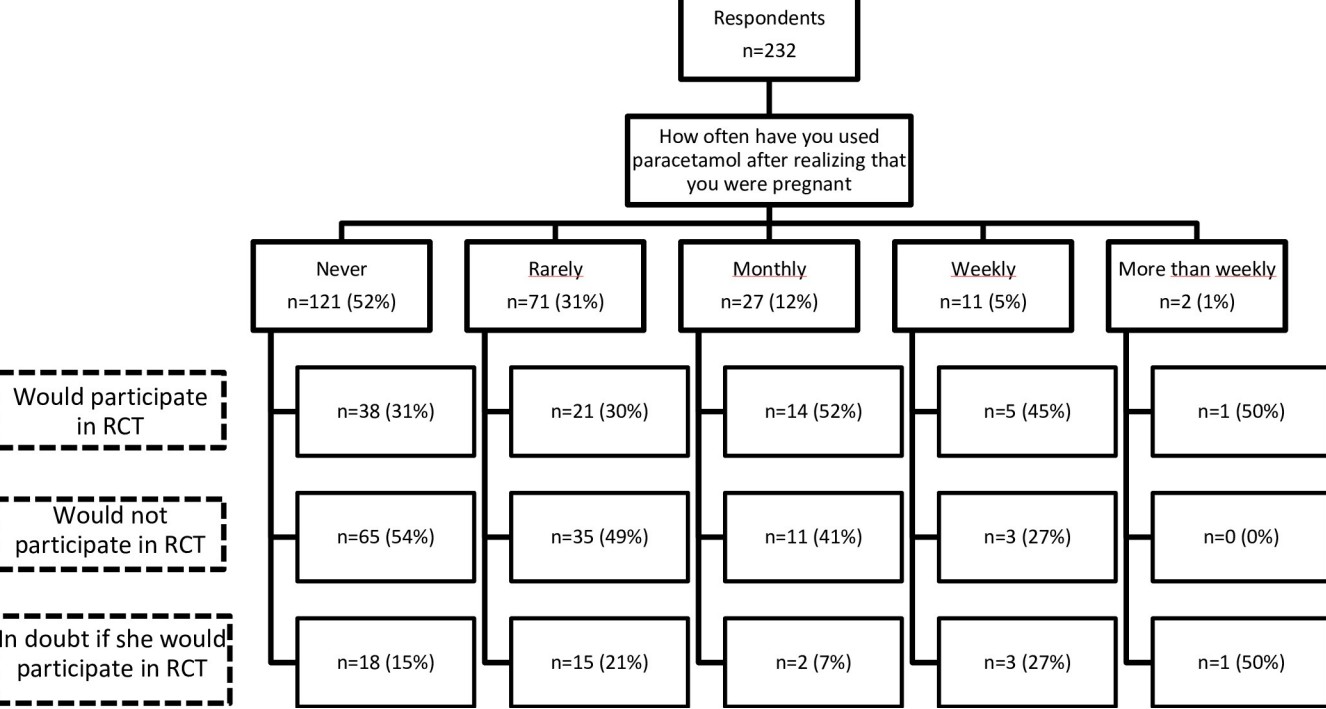

**Fig 1. Flowchart of responses regarding paracetamol consumption during pregnancy and willingness to participate in a randomized clinical trial.**
Numbers are given in absolute numbers, frequencies, and percentages.

who had taken APAP more than weekly (6%), 23% declined to participate in an RCT, leaving 4.3% with a weekly or more than weekly use of APAP during the first trimester of pregnancy who would or would consider participating in an RCT. To recruit 22,000 of these women in Denmark would take 15 years.

The results from the questionnaire also showed that 22.4% of the women would participate in such a study without further compensation, 41% would like a 3<sup>rd</sup>-trimester scan, and the rest marked gift cards as reasonable compensation. When asked how to reach them before the 1<sup>st</sup>-trimester scan, 41% replied through Facebook, 29% through advertisements in established news channels, and 13% through influencers. Moreover, fertility clinics, family doctors, and pharmacies were also mentioned as possible recruitment options.

## Organization of a randomized trial

It is unlikely that the pharmaceutical industry will sponsor a large-scale trial on the side effects of a generic drug. Thus, a trial must be investigator-sponsored and backed by the relevant public and private organizations. The funding must be found elsewhere and is likely to be scarce. Hence the costs need to be minimized and a thoroughly pragmatic approach must be used to reduce efforts as much as possible. Potentially, per patient costs can be very low, as the interventional products (APAP and placebo) are cheap in ingredients and produced at a large scale. To minimize administrative burden, written and oral information about the RCT and retrieval of consent must be done online. The distribution of the interventional products could be done by courier, and all reporting could be web-based–perhaps a phone application (app)–and done by the women themselves. The interface should be very user friendly with only one or to clicks to complete every time medicine is consumed. Of course there is a risk of response bias. However, the risk appears smaller than that of recall bias in studies performed after pregnancy. The primary outcome should be available in the routine health statistics, as Denmark has very comprehensive healthcare registries. There is no apparent ethical hindrance, but the legal obligations of the sponsor may be an issue.

## Possible ways forward

APAP is a possible endocrine disruptor and may have adverse effects on the human urogenital, neurological, and reproductive development when exposed *in utero*. This has been shown in both epidemiological and experimental studies. However, the evidence is for now not strong enough to make the FDA and EMA change their recommendations regarding APAP usage during pregnancy. Our results indicated that a national Danish trial could be completed in 2.5 years. However, we question whether the absolute risk of cryptorchidism is 1% after consuming APAP once or twice during the 1<sup>st</sup> trimester. Restriction to weekly users would extend the trial to 15 years. That does not seem feasible. However, as we see it, there are still other ways forward to organize such a study. It could either be done as collaboration between several countries, as a national study in a country much larger than Denmark, or in a country with a more liberal APAP usage during pregnancy, as it appears to be different across countries with up 65% of the pregnant population in the US consume APAP [18], though with no indication of the frequency of consumption. Access to routinely collected data on surgery for cryptorchidism, however, must be available, to avoid costs of clinical follow-up and loss to follow-up.

The second major issue in performing the proposed clinical trial is how to reach the women at a very early stage in pregnancy. When asked, 41% replied Facebook as an approach. Previous studies have shown that Facebook is a viable option to recruit pregnant women to participate in studies [12–14]. As of ultimo October 2021, the Danish Facebook group with a due date in June of 2022 has a little fewer than 500 members (approximately 10% of all Danish

pregnancies due in June 2022), with increasing numbers in groups closer to their due date. This only applies to the Danish population, and we do not know if pregnant women in other countries have a comparable usage of social media. However, it seems that a combined effort with different recruitment strategies should be used to recruit the women as soon as possible after recognition of the pregnancy.

To address the concerns regarding APAP usage during pregnancy, researchers in maternal and fetal medicine with interest in this matter should explore the possibilities of a randomized clinical trial.

## Supporting information

**S1 Data.**
(XLSX)

**S1 Table.**
(XLSX)

## Acknowledgments

We want to thank Mette Fabricius and the Department of Obstetrics, Slagelse Hospital, as well as Helle Zingenberg and the Department of Obstetrics, Herlev Hospital for the help in collecting data for the qualitative study.

## Author Contributions

**Conceptualization:** David Møbjerg Kristensen, Olav Bjørn Petersen, Gorm Greisen.

**Data curation:** Cathrine Vedel, Ditte Staub Jørgensen.

**Formal analysis:** Cathrine Vedel.

**Investigation:** Cathrine Vedel, Ditte Staub Jørgensen, Gorm Greisen.

**Methodology:** Cathrine Vedel, Ditte Staub Jørgensen, David Møbjerg Kristensen, Olav Bjørn Petersen, Gorm Greisen.

**Project administration:** Cathrine Vedel, Ditte Staub Jørgensen, Olav Bjørn Petersen, Gorm Greisen.

**Supervision:** Olav Bjørn Petersen, Gorm Greisen.

**Writing – original draft:** Cathrine Vedel, Gorm Greisen.

**Writing – review & editing:** Ditte Staub Jørgensen, David Møbjerg Kristensen, Olav Bjørn Petersen.

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
