## [Decision Letter · Decision Letter 0]

13 Apr 2022

PONE-D-21-40201The use of paracetamol during pregnancy: a qualitative study and possible strategies for a clinical trialPLOS ONE

Dear Dr. Vedel,

Thank you for submitting your manuscript to PLOS ONE. After careful consideration, we feel that it has merit but does not fully meet PLOS ONE’s publication criteria as it currently stands. Therefore, we invite you to submit a revised version of the manuscript that addresses the points raised during the review process.

We look forward to receiving your revised manuscript.

Kind regards,

Francesca Crovetto

Academic Editor

PLOS ONE

Reviewers' comments:

Reviewer's Responses to Questions

**Comments to the Author**

1. Is the manuscript technically sound, and do the data support the conclusions?

Reviewer #1: Yes

Reviewer #2: Yes

Reviewer #3: Yes

2. Has the statistical analysis been performed appropriately and rigorously? 

Reviewer #1: Yes

Reviewer #2: I Don't Know

Reviewer #3: Yes

3. Have the authors made all data underlying the findings in their manuscript fully available?

Reviewer #1: Yes

Reviewer #2: Yes

Reviewer #3: Yes

4. Is the manuscript presented in an intelligible fashion and written in standard English?

Reviewer #1: Yes

Reviewer #2: Yes

Reviewer #3: Yes

5. Review Comments to the Author

**Reviewer #1**: First, congratulations for your research project. It is very interesting that you are trying to get into this unknown but very interesting part of fetal medicine. As you said, this generic drugs are not the interest of indutry, what can lead to less research on the topic.

Therefore, I think the aim of your paper is to set the importance of the performance of a RCT and it viability in your population, but it is difficult to get this information in the way you proceed in your paper. This is a descriptive study by a questionnnaire. We should realize this is the type of the study from the beginning. I encourage you to re-focus the main interest of the paper in that, highlighting this aspect:

Lines 42-44. This sentence should go first. It is an introductory statement, not a result. Maybe you could make a categorized summary instead

Lines 82-92: I suggest you to delete this information. It is already explained in detail in the review you cite; there is no need to summarize it here and it confuses the reader of the main focus of the paper. I think your aim is to make a probe of how to design and implement a large RCT in your area, I mean, the results of your questionnaire.

Line 130: Maybe you can add other outcomes and not only cryptorchidism to have a higher incidence of deffects

Also, I have two questions:

Do you suggest to apply this to women with pain or also with fever? Because maybe fever itself can cause some fetal deffects (e.g. cardiac deffects) so you have a selection bias... You should explain that in your paper

How do you recruit women? Those who already took paracetamol before the appointment? Or how do you know which ones will need this drug?

**Reviewer #2:** 

I reported my comments on the manuscript titled “The use of paracetamol during pregnancy: a qualitative study and possible strategies for a clinical trial “.

I think that the topic is interesting. References list appears up to date and appropriate. The manuscript is clear and easy to understand.

Anyway, before the acceptance, I suggest some major revisions.

In the abstract, the sentence “A total of 41% of the women lists Facebook as a possible way to reach them early” appears to be a little out of context. I would suggest eliminating this sentence as you explain the matter more thoroughly later on, in a clearer context.

At line 81, you use the sentence “Hereby follows a brief summary”, which I personally found a bit unnecessary, I would suggest directly explaining the study, as it is implied by the premise.

There is an abbreviation that is used for the first time in the text without explaining its meaning (i.e. ADHD), I suggest checking it. Moreover, I suggest do not use abbreviations in the abstract.

In the paragraph “The limitations of the evidence”, in the first line, you used the expression “by confounding by indication”. I suggest revising this expression.

Regarding your statistical analysis, as I do not possess the adequate skills myself, I have suggested submitting the article to the revision of a statistic specialist, for a more appropriate comment on the numbers you referred.

When you mention the questionnaire, you report only the short version of it, I would suggest reporting the integral questionnaire in a table format.

In the paragraph “Organization of a randomized trial”, you describe the model of an online study, do you think it would be possible to elaborate better on how to maintain high reliability of the data, also considering that any placebo or paracetamol would be self-administered by the women? I am concerned that this distance modality, although inventive, could also hurt the accountability of the data collected.

There are some syntactical mistakes, that I have highlighted here:

At line 105: Studies in human cell… (studies on human cells).

At line 114: …they states… (they state).

At line 121: The golden standard… (the gold standard).

At line 159: … as it was questionnaire (as it was a questionnaire).

Finally, a final suggestion to the authors: evaluating the relationship between the intake of APAP in the first trimester and the presence of malformations is certainly important, however once the effort has been made to recruit women it might be worth thinking about a questionnaire to be administered to those who have taken APAP; similar to the one they use, but with the addition of other targeted questions in order to assess the possible presence of asthma and pre-and post-natal closure of ductus arteriosus, hepatotoxicity and cognitive disorders in the first years of postnatal life (https://www.jstage.jst.go.jp/article/bpb/43/2/43_b19-00722/_article/-char/ja/) DOI https://doi.org/10.1248/bpb.b19-00722

**Reviewer #3:** The article addresses an emerging topic on the safety of a drug that has always been considered safe in pregnancy and suggests how to organize a prospective study, after a rigorous investigation of the Danish.

condition. It is well written. Certainly a prospective study represents a challenge in this case but the group analyzes all the problems and proposes solutions.

6. PLOS authors have the option to publish the peer review history of their article (what does this mean?). If published, this will include your full peer review and any attached files.

Reviewer #1: No

Reviewer #2: **Yes: **Antonio Ragusa

Reviewer #3: No

---

## [Author Response · Author response to Decision Letter 0]

14 Jun 2022

Response to reviewers

Firstly, thank you for taking your time to review our manuscript. It is greatly appreciated. We have gone through all of your concerns, questions, and suggestions and answered to our best.

Reviewer #1: 

First, congratulations for your research project. It is very interesting that you are trying to get into this unknown but very interesting part of fetal medicine. As you said, this generic drugs are not the interest of indutry, what can lead to less research on the topic.

Therefore, I think the aim of your paper is to set the importance of the performance of a RCT and it viability in your population, but it is difficult to get this information in the way you proceed in your paper. This is a descriptive study by a questionnnaire. We should realize this is the type of the study from the beginning. I encourage you to re-focus the main interest of the paper in that, highlighting this aspect:

Lines 42-44. This sentence should go first. It is an introductory statement, not a result. Maybe you could make a categorized summary instead.

We believe the number of included women is important to mention with the results, hence left it as is. We hope that is OK. 

Lines 82-92: I suggest you to delete this information. It is already explained in detail in the review you cite; there is no need to summarize it here and it confuses the reader of the main focus of the paper. I think your aim is to make a probe of how to design and implement a large RCT in your area, I mean, the results of your questionnaire.

We have deleted the section and added a much shortened paragraph instead (l. 80-84).

Line 130: Maybe you can add other outcomes and not only cryptorchidism to have a higher incidence of defects.

Yes, that is a possibility, so we have added suggestions to other outcomes, e.g. hypospadias and adverse neurodevelopmental outcomes (l. 121-125).

Also, I have two questions:

Do you suggest to apply this to women with pain or also with fever? Because maybe fever itself can cause some fetal deffects (e.g. cardiac deffects) so you have a selection bias... You should explain that in your paper.

Yes, we plan to include all applications – also fever. And yes – fever may itself cause congenital malformations, however, the evidence does not quite agree on the matter. But if it does, there is a risk of both harm and benefit using APAP during pregnancy, when applied during hyperthermia. We have elaborated on this (l. 115-118).

How do you recruit women? Those who already took paracetamol before the appointment? Or how do you know which ones will need this drug?

We elaborate on how in l. 120-140, where we have included a sentence regarding early APAP use. Moreover, from l. 240 we discuss the issues with early recruitment.

Reviewer #2: 

In the abstract, the sentence “A total of 41% of the women lists Facebook as a possible way to reach them early” appears to be a little out of context. I would suggest eliminating this sentence as you explain the matter more thoroughly later on, in a clearer context.

We agree and have removed the sentence.

At line 81, you use the sentence “Hereby follows a brief summary”, which I personally found a bit unnecessary, I would suggest directly explaining the study, as it is implied by the premise.

We have deleted the paragraph as also suggested by reviewer 1.

There is an abbreviation that is used for the first time in the text without explaining its meaning (i.e. ADHD), I suggest checking it. Moreover, I suggest do not use abbreviations in the abstract.

It has been removed. We have kept APAP in the abstract, as it is mentioned multiple times.

In the paragraph “The limitations of the evidence”, in the first line, you used the expression “by confounding by indication”. I suggest revising this expression.

We have elaborated the paragraph.

When you mention the questionnaire, you report only the short version of it, I would suggest reporting the integral questionnaire in a table format.

It is actually not a short version, but with comprised versions of each question. We have clarified this in the text.

In the paragraph “Organization of a randomized trial”, you describe the model of an online study, do you think it would be possible to elaborate better on how to maintain high reliability of the data, also considering that any placebo or paracetamol would be self-administered by the women? I am concerned that this distance modality, although inventive, could also hurt the accountability of the data collected.

We have elaborated on this from l. 200.

There are some syntactical mistakes, that I have highlighted here:

At line 105: Studies in human cell… (studies on human cells).

At line 114: …they states… (they state).

At line 121: The golden standard… (the gold standard).

At line 159: … as it was questionnaire (as it was a questionnaire).

Thank you for that. They have all been corrected.

Finally, a final suggestion to the authors: evaluating the relationship between the intake of APAP in the first trimester and the presence of malformations is certainly important, however once the effort has been made to recruit women it might be worth thinking about a questionnaire to be administered to those who have taken APAP; similar to the one they use, but with the addition of other targeted questions in order to assess the possible presence of asthma and pre-and post-natal closure of ductus arteriosus, hepatotoxicity and cognitive disorders in the first years of postnatal life (https://www.jstage.jst.go.jp/article/bpb/43/2/43_b19-00722/_article/-char/ja/) DOI https://doi.org/10.1248/bpb.b19-00722

In Denmark, we have the possibility to perform long-term follow-up through our very extensive registries (l. 204). But it would definitely be interesting to also know what other kinds of medicine the women have taken during pregnancy besides APAP.

---

## [Editor Report · Decision Letter 1]

4 Jul 2022

The use of paracetamol during pregnancy: a qualitative study and possible strategies for a clinical trial

PONE-D-21-40201R1

Dear Dr. Vedel

We’re pleased to inform you that your manuscript has been judged scientifically suitable for publication and will be formally accepted for publication once it meets all outstanding technical requirements.

Kind regards,

Francesca Crovetto

Academic Editor

PLOS ONE

---

## [Editor Report · Acceptance letter]

4 Sep 2022

PONE-D-21-40201R1 

The use of paracetamol during pregnancy: a qualitative study and possible strategies for a clinical trial 

Dear Dr. Vedel:

I'm pleased to inform you that your manuscript has been deemed suitable for publication in PLOS ONE. Congratulations! Your manuscript is now with our production department. 

Kind regards, 

on behalf of

Dr. Francesca Crovetto 

Academic Editor

PLOS ONE